# Diverse Roles of Ceramide in the Progression and Pathogenesis of Alzheimer’s Disease

**DOI:** 10.3390/biomedicines10081956

**Published:** 2022-08-12

**Authors:** Md Riad Chowdhury, Hee Kyung Jin, Jae-sung Bae

**Affiliations:** 1KNU Alzheimer’s Disease Research Institute, Kyungpook National University, Daegu 41566, Korea; 2Department of Physiology, School of Medicine, Kyungpook National University, Daegu 41944, Korea; 3Department of Laboratory Animal Medicine, College of Veterinary Medicine, Kyungpook National University, Daegu 41566, Korea

**Keywords:** Alzheimer’s disease, ceramide, amyloid beta, autophagy, mitochondrial dysfunction, senescence

## Abstract

Alzheimer’s disease (AD) is the most common neurodegenerative disorder, and is associated with several pathophysiological features, including cellular dysfunction, failure of neurotransmission, cognitive impairment, cell death, and other clinical consequences. Advanced research on the pathogenesis of AD has elucidated a mechanistic framework and revealed many therapeutic possibilities. Among the mechanisms, sphingolipids are mentioned as distinctive mediators to be associated with the pathology of AD. Reportedly, alteration in the metabolism of sphingolipids and their metabolites result in the dysfunction of mitochondria, autophagy, amyloid beta regulation, and neuronal homeostasis, which exacerbates AD progression. Considering the importance of sphingolipids, in this review, we discuss the role of ceramide, a bioactive sphingolipid metabolite, in the progression and pathogenesis of AD. Herein, we describe the ceramide synthesis pathway and its involvement in the dysregulation of homeostasis, which eventually leads to AD. Furthermore, this review references different therapeutics proposed to modulate the ceramide pathway to maintain ceramide levels and prevent the disease progression.

## 1. Introduction

Alzheimer’s disease (AD) is defined as progressive cognitive impairment associated with the formation of senile plaques in the cerebral cortex and subcortical gray matter, which includes amyloid beta (Aβ) and neurofibrillary tangles [1]. The World Health Organization has declared AD a “global public health priority” because of the lack of well-established treatments. To date, researchers have proposed various theories and hypotheses regarding the causes and targets of AD. Primarily, dementia is stated as the cause of AD in people over 60 years of age, while approximately 50–75% of patients with dementia develop AD [2]. Furthermore, diabetes, hypertension, and cardiovascular disease have also been identified as risk factors in the progression of AD [3]. Currently, therapeutic interventions are being attempted based on mechanisms that have been proposed in AD pathogenesis. Some of these include antioxidant therapy, NSAIDs, cholinergic replacement therapy, hormone replacement therapy, memantine, and Aβ vaccines [4].

Sphingolipids are abundantly distributed in nerve cell membranes and myelin sheaths of nerve fibers [5]. The bioactive metabolites of sphingolipids, including ceramide, sphingomyelin, sphingosine, and sphingosine-1-phosphate (S1P) are synthesized by a variety of enzymatic biosynthesis pathways [6]. Generally, sphingolipids play a significant role in biological membranes, and their metabolites are involved in the regulation of various cell functions. Several reports have suggested that the metabolism of sphingolipids plays a major role in the pathogenesis of AD, and hence might be considered a potential treatment target [5,7,8]. Additionally, small changes in sphingolipid metabolism may cause remarkable effects in age-related neurodegenerative diseases. In this regard, some studies have demonstrated that the level of ceramides is significantly increased in the brain tissue of patients with AD compared to controls, whereas sphingomyelin and sphingosine-1-phosphate are decreased [5,6,7]. In addition, abnormal expression of enzymes in the sphingolipid synthesis pathway have also been described previously [9]. Moreover, plasma sphingolipids are considered as potential biomarkers of neurodegenerative diseases [10]. In particular, one study revealed that the levels of sphingomyelin and ceramide were altered in the plasma of patients with AD as compared to the control group [11]. The study was further supported by another report, in which an increase in sphingomyelin levels was detected in the cerebrospinal fluid of patients with AD [12].

Different bioactive molecules of sphingolipids exhibit several regulatory functions in cellular and tissue homeostasis, particularly senescence, cell cycle, migration, proliferation, autophagy, inflammation, and immune responses [6]. While these mediators are abundant in the central nervous system, any dysfunction of sphingolipids may contribute to neurodegenerative disorders. Especially, ceramide is one of the simplest sphingolipids which is widely distributed in animal tissues, while other sphingolipids are derivatives of ceramide [13]. Therefore, in this review, we aimed to discuss the impact of ceramide, a bioactive metabolite of sphingolipids, on the progression and pathogenesis of AD. Moreover, this review mentioned a list of therapeutics that are suggested to mediate ceramide synthesis pathway and prevent the disease progression.

## 2. Pathogenesis of Alzheimer’s Disease

Although the pathogenesis of AD is not clear, it is known that the disease involves several factors, including neurotransmitters, immunity, hereditary and environmental factors [14]. Currently, researchers have proposed different theories regarding AD, namely, the amyloid beta theory, tau theory, oxidative stress, mitochondrial dysfunction, inflammation, autophagy dysfunction, and nerve and blood vessel theory [2,14]. Among these, the concept of Aβ-containing senile plaque formation has previously been described by many researchers. Aβ is not deposited in the brain under normal conditions, as it is generally cleared by the homeostatic clearance mechanism. However, an imbalance in the rate of Aβ production and clearance results in the formation of amyloid plaques. A previous study revealed that Aβ deposition in the brain is associated with cognitive impairment in the elderly [15]. Therefore, preventing Aβ sedimentation could be a potential method to ameliorate AD. In contrast, the development of neurofibrillary tangles by hyperphosphorylated tau protein is a fundamental neuropathological marker of AD [16]. Abundantly distributed in neurons, the tau protein plays a role in maintaining the stability of cytoskeletal microtubules and axonal transport [17]. Nevertheless, due to chemical dysfunction in AD, tau tends to separate from microtubules and attach to other tau molecules to form tangles, blocking the transport of neurons [18]. Consequently, obstruction of synaptic communication between neurons promotes the progression and pathogenesis of AD. With regards to the synapses, previous report confirmed that during AD the synaptic plasticity changes and causes reduced synapses in the brain [14]. Collectively, the deposition of Aβ, aggregation of tau protein, and synaptic loss cause neuronal injury followed by neuronal death and contribute to cognitive impairment.

Chemokines are a group of secreted proteins mostly known as a regulator of cell migration, especially leukocytes. The chemokines are associated with AD and considered as a key factor in the pathology of AD because of their involvement in the regulation of inflammatory or glial cells. A previous report suggested that the level of chemokines in the brain, cerebrospinal fluid, and serum constantly fluctuate in patients with AD [14]. Due to their role in regulating proinflammatory and anti-inflammatory properties, chemokines cause neuroinflammation in the AD brain, subsequently leading to neuronal death. Microglia, a type of glial cell regulated by chemokines function as an immune defense in the central nervous system. However, previous study has reported that microglia are unable to clear waste and toxins from the brain, and their accumulation can cause chronic neuroinflammation [18]. In addition, brain cells are dependent on mitochondrial oxidative phosphorylation for their energy. However, gene set enrichment analysis has shown significant downregulation of mitochondrial oxidative phosphorylation and interruption of the mitochondrial import pathways in AD [19]. As such, mitochondrial dysfunction may also be correlated with the pathogenesis of AD. Besides, autophagy is an important cell survival mechanism that facilitates bioenergetic homeostasis. Researchers have further discussed the involvement of autophagy dysfunction in the pathogenesis of AD, as increased autophagy causes the accumulation of Aβ in the brain [20].

Another key regulator of brain function is lipid that has been increasingly involved in AD. It was previously observed that in AD patients, heterogeneous changes in lipid metabolism occurs in different regions of the brain [21]. Especially, lipid composition in the neuron is able to regulate the activity of membrane-bound proteins including BACE1, APP and presenilin thereby adjust the amyloid beta levels [21]. Accordingly, Grimm et al. reported that the level of cholesterol and sphingomyelin in the membrane can regulate the γ-secretase activity [22]. Cholesterol also accumulated in nerve terminals and Aβ plaques in human AD brain and APP transgenic mice [23]. Furthermore, production and accumulation of Aβ is stimulated due the elevation in cellular cholesterol level [24]. On the other hand, Aβ is reported to be modulating the phospholipase activity and exert its cytotoxic effects by disturbing the cell membranes [21]. Another lipid named ganglioside GM1 can regulate the pathogenic potential of Aβ by controlling susceptibility to aggregate [25]. Therefore, altered lipid metabolism could be an important contributor in the pathogenesis of AD.

In short, the association of amyloid beta toxicity, tau protein aggregation, synaptic damage, mitochondrial dysfunction, oxidative damage, neuroinflammation, and autophagy dysfunction complicates the AD pathogenesis. 

## 3. Ceramide Synthesis Pathways

### 3.1. De Novo Synthesis

De novo synthesis of ceramide occurs at the cytosolic leaflet of the endoplasmic reticulum. The reaction is primarily initiated by the condensation of cytosolic serine and palmitoyl CoA by serine palmitoyltransferase to produce 3-ketodihydrosphingosine [26]. In turn, 3-ketodihydrosphingosine is reduced to dihydrosphingosine by 3-ketodihydrosphingosine reductase, followed by the production of dihydroceramide by dihydroceramide synthase [27]. Subsequently, dihydroceramide is catalyzed by dihydroceramide desaturase to form ceramide, which is transported to the Golgi apparatus (Figure 1) [28].

### 3.2. Salvage Pathway

Sphingomyelin and glucosylceramide are produced by the degradation of sphingolipids and glycosphingolipids in acidic subcellular compartments, lysosomes, and/or late endosomes [29]. Subsequently, sphingomyelin is cleaved to produce ceramide through the action of acid sphingomyelinase (ASM) [30]. In contrast, glucosylceramide is converted into ceramide by the acid β-glucosidase 1 enzyme [29]. Furthermore, lysosomal ceramides produced by these processes are deacetylated by acid ceramidase to produce sphingosine, which can eventually form sphingosine-1-phosphate or ceramide [31]. Ceramide also acts as a substrate to produce sphingosine, a product of sphingolipid catabolism [32]. Thus, this pathway of sphingosine recycling is termed the “salvage pathway,” and is responsible for regulating the formation of ceramide.

### 3.3. Sphingomyelin Hydrolysis

Sphingomyelin (SM) is an abundant sphingolipid in the plasma membrane and can be degraded into ceramide and phosphocholine. There are three types of SMase that catalyzes sphingomyelin namely, acid sphingomyelinase, neutral sphingomyelinase and alkaline while SM hydrolysis involves catalysis by neutral sphingomyelinase [27]. Additionally, a previous report suggested that in some cells, ionizing radiation activates sphingomyelinase in the cell membrane to produce ceramide [33].

## 4. The Role of Ceramide in AD Cell Lines

### 4.1. Amyloid Beta Plaques

Amyloid beta peptide (Aβ) is known to originate from the transmembrane protein amyloid beta precursor protein (APP). APP is cleaved by a beta-secretase enzyme called beta-site APP cleaving enzyme 1 (BACE1), releasing the C99 fragment of APP, which eventually results in the production of the Aβ peptide. Subsequently, these peptides are secreted into the extracellular region of the brain and transported by the cerebrospinal fluid. Aβ exhibits several beneficial roles in human physiology, including tumor suppression, regulation of synaptic function, and recovery from BBB (blood brain barrier) leakage and brain injury [34]. However, abnormal accumulation of Aβ peptide in the brain has been reported to be the main pathogenic cause of disruption of the neuronal cell function which triggers Alzheimer’s disease [34]. Moreover, it has been hypothesized that during AD, the level of secreted Aβ peptide is gradually elevated in the extracellular space and forms insoluble amyloid fibrils that are resistant to degradation [35]. Interestingly, a previous report suggested that the level of ceramides in the brains of patients with AD is increased three-fold compared to age-matched controls [36]. In addition, many studies have revealed the progressive elevation of ceramide levels throughout aging in both cultured cells and the whole brain samples [37]. Thus, ceramide might be capable of regulating the production of Aβ and increasing the risk of AD. In particular, a study of C6-ceramide reported that this cell-permeable analog upregulated the generation of Aβ in Chinese hamster ovary cells [38]. Furthermore, the authors demonstrated that C6-ceramide promotes the β-cleavage of APP and regulates the molecular stability of BACE1, thereby increasing the rate of Aβ biosynthesis [38] (Figure 2). Accordingly, the action of ceramide on BACE1 might be a strong pathway through which Aβ generation increases and forms Aβ plaques during AD.

### 4.2. Mitochondrial Dysfunction

In neurons, the normal functional roles of mitochondria include regulating calcium homeostasis, neurotransmission, membrane excitability management, and plasticity. In addition, myriad neuronal cell functions, including calcium buffering and ATP production, are also dependent on mitochondria [39]. Thus, defective mitochondrial function is anticipated to cause cellular dysfunction and contribute to disease progression. Similarly, prolonged mitochondrial damage induces dysregulation of energy metabolism, which can cause reduced energy (ATP) production, calcium buffering, and increased reactive oxygen species (ROS) [40]. Previously, ceramide is reported to modulate mitochondrial function and oxidative phosphorylation wherein, certain ceramides are involved with reduced mitochondrial respiratory chain (MRC) activity, elevated ROS production, oxidative stress and decreased mitochondrial membrane potential [41]. Many studies have suggested that ceramides act locally on the mitochondria rather than in any other cell organelle [41,42,43]. Consequently, ceramide may directly contribute to the generation of free radicals in the mitochondria, leading to mitochondrial dysfunction and apoptosis [42]. This was further confirmed by another study which reported that mitochondrial dysregulation by ceramide may induce oxidative stress and activate the apoptotic indicator poly (ADP-ribose) polymerase-1 (PARP-1) [41]. Moreover, ceramide-induced oxidative stress elevates the oxidase activity of NADPH and produces H_2_O_2_ which causes neuronal damage [44] (Figure 2). In contrast, the central nervous system is reported to be affected most frequently by mitochondrial damage, which eventually leads to cognitive decline and dementia [43]. A recent investigation showed that disrupted mitochondrial function impairs neuronal stem cell self-renewal and causes defects in neurogenesis, differentiation, and neuronal survival [45]. Therefore, the irregular action of ceramide on mitochondria might be a possible mechanism of mitochondrial dysregulation, which eventually contributes to the progression and pathogenesis of AD.

### 4.3. Senescence

Senescence is an irreversible cellular process in which cells lose the ability to proliferate. Many factors are involved in the initiation of cellular senescence, including organelle stress, DNA damage, oncogene activation, and telomere dysfunction [46]. In addition, many studies have suggested that senescence is regulated by different sphingolipids [37,47,48]. Particularly, one study reported an increase in the level of ceramides in old senescent WI-38 human fibroblasts as compared to young fibroblasts [49]. Their investigation showed that ceramide significantly inhibited the growth of WI-38 human diploid fibroblasts. Similar cell growth arrest and differentiation were observed in senescent cells by another study [48]. Ceramide has also been reported to activate PP2A (protein phosphatase 2A-like protein phosphatase), a family of enzymes that regulate cellular processes and signal transduction pathways [50]. Study have suggested that C6-ceramide directly activates PP2A, which causes subsequent inhibition of cyclin-dependent kinase-2 and leads to cell cycle arrest [51] (Figure 2). Notably, ceramide might also affect tau hyperphosphorylation through the modulation of PP2A activity and contribute to the progression of AD [42]. In addition, a hydrolase enzyme, senescence-associated beta-galactosidase (SA-β-gal), allows the identification of cell senescence in mammalian tissue. One report demonstrated that C6-ceramide-induced increased expression of SA-β-gal in WI-38 cells in a time-and dose-dependent manner [37]. Collectively, these results support the significant contribution of ceramide to cellular senescence through different mechanisms.

Senescence has been associated with several pathologies, including cancer, type-2 diabetes, fibrosis, atherosclerosis, and Alzheimer’s disease [47]. Besides, different studies have described the role of senescence in the pathology of AD. Features of specific cells during AD exhibit features indicative of cells that undergoes senescence. For example, telomeric alterations and hyperproliferation-mediated DNA damage in cellular senescence have been observed in AD as well [52]. Particularly, one study demonstrated the correlation between AD and DNA double strand breaks where the authors have observed an increased γ-H2A.X (Ser139) positive neurons and GFAP positive astrocytes in the hippocampus of human AD postmortem brains compared to non-AD brains [53]. These results indicate the accumulation of DNA double strand breaks during AD condition which activates DDR (DNA damage response) and lead to senescence [53]. A previous study suggested that the removal of senescent cells in aged mice elevates their lifespan and decreases the likelihood of age-related diseases, including AD [54]. Similarly, another study compared the transcriptome in young and old mice hippocampi and reported that senescence markers were accumulated in microglia [55]. Moreover, the elimination of senescent microglia significantly decreased neuroinflammation and restrains cognitive decline [55]. This evidence supports the possibility that cellular senescence by ceramide increases the progression of AD.

### 4.4. Autophagy Dysfunction

Autophagy is an intracellular bulk degradation process that targets the cytosolic content and organelles in mammalian cells. Based on differences in functions and mechanisms, three types of autophagy have been described: microautophagy, macroautophagy, and chaperone-mediated autophagy [56]. The biogenesis of autophagy involves several steps, including phagophore membrane isolation, autophagosome maturation, prolongation of phagophore, and cytoplasmic content engulfment [57]. Although autophagy induction is largely triggered by cellular starvation, ceramide has also been reported to be involved in autophagy activation [58]. Several studies have confirmed that ceramide induces lethal autophagy. Scarlatti et al. demonstrated that C2 ceramide increased the expression of Beclin-1 and induced lethal autophagy [59]. Similarly, Qian et al. showed that upregulation of Beclin-1 contributed to lethal autophagy, which could be prevented by the administration of the autophagy inhibitor 3-MA [60]. In another study, the mechanism of lethal autophagy was investigated, and it was found that the c-Jun protein was activated through JNK signaling by ceramide and upregulated the expression of Beclin-1 (Figure 2) [61]. Furthermore, ceramide-induced autophagic cell death was intercepted through the inhibition of JNK activity by the administration of SP600125 [61].

Autophagic dysfunction has been reported to contribute to neurodegenerative disorders by triggering neuronal cell death [62]. A previous report suggested that autophagy dysfunction significantly contributes to the pathophysiology of AD [63]. One study by Nixon et al. found a significant accumulation of autophagosomes in the frontal cortex of AD patients in comparison to controls [64]. In addition, Yu et al. demonstrated that APP residing in autophagic vacuoles can produce Aβ, which may act as a source of Aβ production in AD [65]. Interestingly, the production and release of Aβ, as well as the formation of plaques, are dependent on the induction of autophagy [66]. To further understand the role of autophagy in Aβ secretion, Nilsson et al. generated autophagy-deficient APP transgenic mice, and observed that the secretion of Aβ decreased by 90% [67]. Moreover, autophagy-lysosome dysfunction is reported to stimulate tau formation and tau species accumulation which further contributes to the pathology of AD [68]. These reports confirm a strong role of ceramide in autophagy dysfunction to exacerbate the pathogenesis of AD. 

## 5. Role of Ceramide in Plasma on AD

Ceramides in the blood circulation are transported by lipoproteins. Dysregulation in the plasma concentration of ceramide is reported in many disorders. Likewise, ceramide dysregulation has previously been reported to be detected in the circulation of patients with AD in several studies. Accordingly, it has been hypothesized that plasma ceramide may be correlated with AD, and thus it might act as a potential target to prevent the disease progression. Particularly, one report suggested that elevated plasma ceramide levels are associated with the increased risk of cognitive impairment and AD among cognitively normal individual [69]. Besides, ceramides are also associated with platelet activation and endothelial dysfunction through the detachment of NO signaling pathway [70]. It was previously reported that platelets are the major source of both APP and Aβ in human blood [71]. Therefore, it is possible that the activation of platelet by plasma ceramide might contribute to the pathogenesis of AD by elevating the level of Aβ. Various reports have proposed that the pathogenesis of systemic inflammation and insulin resistance is associated with plasma ceramide [69,72,73]. In addition, plasma ceramides are linked with proinflammatory cytokines in patients with type-2 diabetes, obesity, and cardiovascular disease [73]. For example, the production and release of proinflammatory cytokines are stimulated by LDL-ceramide following accumulation of ceramide linked to JNK and NF-κβ signaling [74]. Consequently, elevated levels of LDL-ceramide exert proinflammatory effects on macrophages and promotes inflammation in obesity [74]. These cytokines cross the blood-brain barrier and activate inflammatory pathways in the hypothalamus to cause dysregulation in brain homeostasis [75]. Several reports have suggested that patients with type-2 diabetes develop cognitive impairment, wherein insulin resistance is initiated in the brains of patients with AD [73,76,77].

Moreover, a report have demonstrated the relationship between ceramide and depression, where the ceramide pathway is considered as a target for antidepressants [78]. Xing et al. showed that plasma ceramide levels were related to depression in moderate-to-severe AD [79]. Another similar study reported that patients with anxiety and major depressive disorder progress hippocampal atrophy which is supported by a report where an increase in the level of proinflammatory cytokines tumor necrosis factor-ɑ, interlukin-6 and interlukin-1 is seen in MDD patients [80]. Collectively, ceramide is responsible for initiating depression while major depressive disorder increases the risk of developing AD and other forms of dementia.

Overall, ceramide has been correlated with neurodegenerative diseases, and alterations in plasma ceramide levels have been identified in the cerebrospinal fluid of patients with AD [73]. In a recent report, Mielke et al. showed that an increased level of serum Cer16:0 and Cer24:0 is associated with a higher risk of developing AD [10]. In addition, clinical studies and laboratory and animal investigations have shown that an imbalance in plasma sphingomyelin and ceramide levels causes the progression of AD through amyloid beta formation and subsequent neurodegeneration [10].

## 6. Therapeutics Targeting Ceramide Biosynthesis

Currently, the level of ceramide can be modulated using various therapeutic drugs. When designing these agents, researchers have targeted different enzymes that contribute to the ceramide synthesis pathway. In terms of de novo synthesis of ceramide, myriocin and fumifungin, which are structurally similar to sphingofungins, have been reported to inhibit the enzyme serine palmitoyltransferase [81] (Table 1). Mandala et al. investigated the mechanism of action of the antifungal agent lipoxamycin and revealed that lipoxamycin and hydroxylipoxamycin could potentially inhibit serine palmitoyltransferase [82]. Another compound, fumonisin B1, which is also structurally similar to that of sphingosine, has been reported to inhibit ceramide synthase in the de novo synthesis pathway [5]. In addition, dihydroceramide desaturase 1 (DES1) plays a major role in the final step of de novo ceramide biosynthesis. Triola et al. reported for the first time an effective synthesized DES1 inhibitor, GT11 (C8-cyclopropenylceramide), which is a ceramide derivative [83]. A different sphingolipid analog, XM462 (5-thiadihydroceramide), was also reported as a DES1 inhibitor to prevent ceramide biosynthesis [84]. In addition, some non-sphingolipid analogs, including fenretinide, SKI II, celecoxib, resveratrol, curcumin, and Δ9- tetrahydrocannabinol (THC), have been reported to exhibit inhibitory activity against DES1 [85].

Neutral sphingomyelinase (N-sphingomyelinase) is another enzyme, which is required for the hydrolysis of sphingomyelin to synthesize ceramide. Accordingly, N-sphingomyelinase inhibition is an effective approach to regulate ceramide levels. A previous report suggested that an isolated compound, scyphostatin, has a structural resemblance to ceramide and may inhibit N-sphingomyelinase [88]. Moreover, some other N-sphingomyelinase inhibitor analogs have been developed, namely spiroexpoxide, [89] manumycin A, [90] alutenusin, [91] and ubiquinol [92].

In addition, the hydrolysis of sphingomyelin is catalyzed by ASM (acid sphingomyelinase), indicating another target for the regulation of ceramide biosynthesis. To date, different types of ASM inhibitors have been reported, including natural, non-natural, physiological, and functional inhibitors. Besides, several research groups have reported some direct inhibitors of ASM including KARI 201 [93], ARC39 [94], AD2765 [95], and SMA-7 [96]. In addition, α-mangostin [97], cowanol, and cowanin [85] are natural ASM inhibitors, while various physiological inhibitors, including L-αphosphatidyl-D-myoinositol-3,5-biphosphate [98] and phosphatidyl-myo-inositol-3,4,5-triphosphate [99] also inhibits ASM. Indirect inhibitors of ASM, which act as functional inhibitors of acid sphingomyelinase (FIASMAs) are also available. FIASMAs consist of compounds such as fluoxetine, dextromethorphan, maprotilin, orphenadrine, nortriptyline, flupromazine, and sertraline [100]. These agents have been reported to inhibit ASM in the salvage pathway of ceramide synthesis, thereby reducing the production of ceramide as a therapeutic agent for neurological disorders.

Among these drugs many have undergone different phases of clinical trials including Fenretinide, Resveratrol, Ubiquinol, Fluoxetine, Nortriptyline and Sertraline for their investigation against recurrent neuroblastoma, cognitive change, sepsis, major depressive disorder, depression, Parkinson disease and anxiety respectively.

## 7. Conclusions and Future Perspective

As the population continues to age, the prevalence of AD is also increasing. To date, researchers have proposed several theories to explain the pathogenesis of AD. Based on these mechanisms various therapeutics are developed and currently available to prevent the disease progression. Among the mechanisms of AD pathology, most of the previous reports have demonstrated an association of sphingolipid and its bioactive metabolites in the progression and pathogenesis of AD. In this review, we emphasize the role of ceramide, an important bioactive lipid in the sphingolipid family, in AD pathology. Ceramide is associated with different pathological states, including neurodegeneration, diabetes, obesity, and inflammation. As described above, ceramide levels in the cell and plasma significantly control several characteristics of pathological changes in AD, including Aβ plaques, mitochondrial dysfunction, senescence, autophagy dysfunction, platelet activation and endothelial dysfunction. Moreover, unlike other sphingolipids, ceramide directly participates in initiating the abovementioned pathological changes to cause AD. Hence, ceramide biosynthesis pathways and the participating enzymes could be potential targets to regulate ceramide levels and prevent disease progression. Several studies have suggested the use of different types of drugs to inhibit specific enzymes in the ceramide synthesis pathway. Consequently, the modulation of ceramide synthesis pathway significantly controlled the level of ceramide and decreased AD progression. Nevertheless, owing to the lack of knowledge regarding their safety and efficacy, the search for potential targets and remedies remains a major point of attention.

## Figures and Tables

**Figure 1 biomedicines-10-01956-f001:**
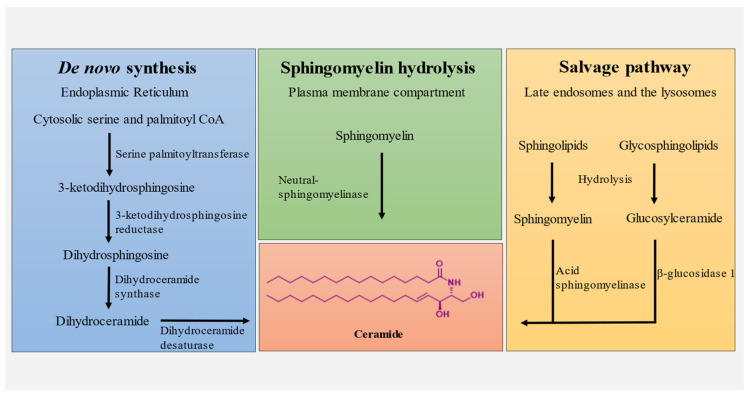
Ceramide synthesis pathways; de novo synthesis, sphingomyelin hydrolysis and salvage pathway.

**Figure 2 biomedicines-10-01956-f002:**
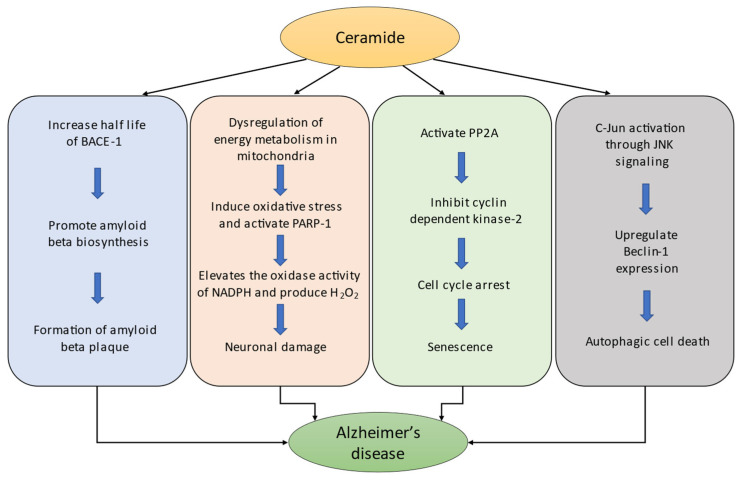
Mechanistic role of ceramide in the pathogenesis of AD.

**Table 1 biomedicines-10-01956-t001:** List of drugs targeting different ceramide biosynthesis pathways.

Pathway	Targets	Drugs	Reference
De novo synthesis	Serine palmitoyltransferase	Myriocin Fumifungin Lipoxamycin Hydroxylipoxamycin	[81] [86] [82]
	Ceramide synthase	Fumonisin B1	[5,87]
	Dihydroceramide desaturase 1	GT11 (C8-cyclopropenylceramide) XM462 (5-thiadihydroceramide) Fenretinide SKI II Celecoxib Resveratrol Curcumin Δ9-tetrahydrocannabinol (THC)	[83] [84] [85]
Sphingomyelin hydrolysis	Neutral sphingomyelinase	Scyphostatin Spiroexpoxide Manumycin A Alutenusin Ubiquinol	[88] [89] [90] [91] [92]
Salvage pathway	Acid sphingomyelinase	KARI 201 ARC39 AD2765 SMA-7 α-mangostin Cowanol and Cowanin L-αphosphatidyl-D-myoinositol-3,5-biphosphate Phosphatidyl-myo-inositol-3,4,5-triphosphate Fluoxetine Dextromethorphan Maprotilin Orphenadrine Nortriptyline Triflupromazine Sertraline	[93] [94] [95] [96] [97] [85] [98] [99] [100]

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
