# Peer review of "Diverse Roles of Ceramide in the Progression and Pathogenesis of Alzheimer’s Disease"

_biomedicines, 2022, doi:10.3390/biomedicines10081956_

Round 1

Reviewer 1 Report

In this review manuscript, authors summarize and discuss about the current knowledges for the role of ceramide in the progression and pathogenesis of Alzheimer’s disease (AD). As sphingolipids play an important role in the central nervous system, their focus is timely and significant. However, there are several points which should be further addressed.

1. There is no sufficient discussion for the role of ceramide in other neurodegenerative diseases. Dysregulations of ceramide metabolism might not be a specific phenotype to AD.

2. Altered lipid metabolism is involved in AD pathogenesis. It is unclear if ceramide predominantly mediates the pathogenic mechanism compared to cholesterol and other phospholipids in AD. Although some lipidomics studies in human AD plasma, CSF and brain samples were published, there is no sufficient discussion regarding how lipidomics studies support contributions of ceramide to AD pathogenesis

3. The evidence from human postmortem brains should be discussed in more detail by creating a new section.

4. There is no clear discussion for the relationship between ceramide and tauopathy. Is ceramide specifically involved in amyloid pathology?

5. While APOE4 is the strongest genetic risk factor for AD, it is not sufficiently described regarding how APOE influences sphingolipid metabolism.

Author Response

Comments and Suggestions for Authors

In this review manuscript, authors summarize and discuss about the current knowledges for the role of ceramide in the progression and pathogenesis of Alzheimer’s disease (AD). As sphingolipids play an important role in the central nervous system, their focus is timely and significant. However, there are several points which should be further addressed.

  1. There is no sufficient discussion for the role of ceramide in other neurodegenerative diseases. Dysregulations of ceramide metabolism might not be a specific phenotype to AD.

We would like to thank the reviewer for evaluating the manuscript. Although the dysregulation of ceramide metabolism is correlated with different neurodegenerative diseases, our review mainly focused on the mechanisms through which ceramide contributes to the pathogenesis of AD solely.

  1. Altered lipid metabolism is involved in AD pathogenesis. It is unclear if ceramide predominantly mediates the pathogenic mechanism compared to cholesterol and other phospholipids in AD. Although some lipidomics studies in human AD plasma, CSF and brain samples were published, there is no sufficient discussion regarding how lipidomics studies support contributions of ceramide to AD pathogenesis

We would like to thank the reviewer for this point. We agree with the reviewer’s comment as dysregulated metabolism of lipid is associated with AD pathogenesis. Since our study did not discuss all the lipids, we specifically highlighted sphingolipid as an important contributor in the pathology of AD.

  1. The evidence from human postmortem brains should be discussed in more detail by creating a new section.

We would like to thank the reviewer for this important point. According to the suggestion of the reviewer, in section 4.3, we have included the detailed explanation of the evidence from human postmortem brain.

  1. There is no clear discussion for the relationship between ceramide and tauopathy. Is ceramide specifically involved in amyloid pathology?

We would like to thank the reviewer for this comment. Our review discussed the major role played by ceramide in the pathogenesis of AD. Although the role of ceramide in tau phosphorylation is not as crucial as amyloid pathology, we further added a reference correlating between ceramide and tau hyperphosphorylation in section 4.3.

  1. While APOE4 is the strongest genetic risk factor for AD, it is not sufficiently described regarding how APOE influences sphingolipid metabolism.

We would like to thank the reviewer for this comment concerning our manuscript. The focus of our review is to discuss about the consequences of dysregulation in the metabolism of sphingolipid especially in term of AD rather than the causes of dysregulation. Therefore, we exclusively highlighted the role of ceramide in the pathogenesis of AD.

Reviewer 2 Report

This review paper well illustrates and summarizes the potential roles of ceramide in the progression of Alzheimer’s disease. The authors further provide insight into how current knowledge can be applied in near future for developing efficient therapeutics against the disease. I have several minor suggestions to improve the rigor of the manuscript.

As the authors described, other sphingolipids including sphingomyelin and sphingosine-1-phosphate are also reported to be associated with AD progression. It is not clearly stated in the text why the authors focused on ceramide / why the dysregulation of ceramide than other sphingolipids can be more detrimental.  

Regarding Table-1, have any of these drugs been tested in clinical trials?

It would be great if the authors elaborate on the advantage of targeting ceramide over the current therapeutical approaches.   

Author Response

Comments and Suggestions for Authors

This review paper well illustrates and summarizes the potential roles of ceramide in the progression of Alzheimer’s disease. The authors further provide insight into how current knowledge can be applied in near future for developing efficient therapeutics against the disease. I have several minor suggestions to improve the rigor of the manuscript.

  1. As the authors described, other sphingolipids including sphingomyelin and sphingosine-1-phosphate are also reported to be associated with AD progression. It is not clearly stated in the text why the authors focused on ceramide / why the dysregulation of ceramide than other sphingolipids can be more detrimental.  

We would like to thank the reviewer for this important suggestion. We have revised the manuscript and added the purpose of focusing on ceramide in section 1.

2. Regarding Table-1, have any of these drugs been tested in clinical trials?

Thank you for your impactful question. We have added another paragraph in section 6 which contains the drug that have undergone clinical trials.

3. It would be great if the authors elaborate on the advantage of targeting ceramide over the current therapeutical approaches.   

Thank you for your valuable suggestion. We have added the advantage of considering ceramide as a potential therapeutic target (In section 7).

Round 2

Reviewer 1 Report

This reviewer still recommends to describe altered lipid metabolism in AD pathogenesis. Although ceramide might play a critical role in AD, the review manuscript somehow lacks the balanced view.

Author Response

This reviewer still recommends to describe altered lipid metabolism in AD pathogenesis. Although ceramide might play a critical role in AD, the review manuscript somehow lacks the balanced view.

: We would like to thank the reviewer for this suggestion. According to the recommendation, we have included a separate paragraph in section 2 discussing the role of altered lipid metabolism in the pathogenesis of AD.